# Expression of *HOXA10* Gene in Women with Endometriosis: A Systematic Review

**DOI:** 10.3390/ijms241612869

**Published:** 2023-08-17

**Authors:** Nurunnajah Lazim, Marjanu Hikmah Elias, Zulazmi Sutaji, Abdul Kadir Abdul Karim, Mohammad Azrai Abu, Azizah Ugusman, Saiful Effendi Syafruddin, Mohd Helmy Mokhtar, Mohd Faizal Ahmad

**Affiliations:** 1Advanced Reproductive Centre (ARC) HCTM UKM, Department of Obstetrics & Gynecology, Faculty of Medicine, National University of Malaysia, Jalan Yaacob Latiff, Bandar Tun Razak, Kuala Lumpur 56000, Malaysia; misznajah@gmail.com (N.L.); marjanuhikmah@usim.edu.my (M.H.E.); zulazmisutaji@yahoo.com (Z.S.); azraiabu1983@gmail.com (M.A.A.); abdulkadirabdulkarim@yahoo.com (A.K.A.K.); 2Faculty of Medicine & Health Sciences, Universiti Sains Islam Malaysia, Nilai 71800, Negeri Sembilan, Malaysia; 3Department of Physiology, Faculty of Medicine, National University of Malaysia, Jalan Yaacob Latiff, Bandar Tun Razak, Kuala Lumpur 56000, Malaysia; dr.azizah@ppukm.ukm.edu.my (A.U.); helmy@ukm.edu.my (M.H.M.); 4Medical Molecular Biology Institute, National University of Malaysia, Jalan Yaacob Latiff, Bandar Tun Razak, Kuala Lumpur 56000, Malaysia; effendisy@ppukm.ukm.edu.my

**Keywords:** endometriosis, endometrium, gene, gene expression, *HOX*, *HOXA10*

## Abstract

The homeobox A10 (*HOXA10*) gene is known to be related to endometriosis; however, due to a lack of knowledge/evidence in the pathogenesis of endometriosis, the mechanisms that link *HOXA10* to endometriosis still need to be clarified. This review addresses the difference in the expression of the *HOXA10* gene in endometriotic women versus non-endometriotic women across populations by country and discusses its influences on women’s fertility. An organized search of electronic databases was conducted in Scopus, ScienceDirect, PubMed, and Web of Science. The keywords used were (HOXA10 OR “homeobox A10” OR PL OR HOX1 OR HOX1H OR HOX1.8) AND (“gene expression”) AND (endometriosis). The initial search resulted in 623 articles, 10 of which were included in this review. All ten papers included in this study were rated fair in terms of the quality of the studies conducted. The expression of the *HOXA10* gene was found to be downregulated in most studies. However, one study provided evidence of the downregulation and upregulation of *HOXA10* gene expression due to the localization of endometriotic lesions. Measuring the expression of the *HOXA10* gene in women is clinically essential to predicting endometriosis, endometrial receptivity, and the development of pinopodes in the endometrium during the luteal phase.

## 1. Introduction

Homeobox genes were initially discovered in the fruit fly *Drosophila* due to a mutation that causes a segment of the fruit fly’s body to transform into another similar segment [1]. Hox genes are divided into four clusters, clusters A-D, which consist of *HOXA*, *HOXB*, *HOXC*, and *HOXD* [2]. Hox genes are involved in the development of the female reproductive tract during embryogenesis. HOX A gene clusters are regionally expressed during embryonic development in mammals such as mice and humans (Figure 1). Specifically, the expression of *HOXA10/Hoxa10* is observed in the developing uterus, while *Hoxa11* is observed in both the developing uterus and the cervix [3]. *HOXA10/Hoxa10* plays a vital role in regulating the expression of a factor that affects the capabilities of preimplantation embryos. In mice, disrupting *Hoxa10* genes resulted in fertility problems [4]. The expression of endometrial *HOXA10* is influenced by sex steroid hormones such as progesterone and estrogen. This discovery shows that *HOXA10* expression can be affected by the stage of the menstrual cycle. *HOXA10* is found to be expressed in the first and second halves of the proliferative phase and the first third of the secretory phase without significant differences. However, a sudden increase in *HOXA10* was observed during the mid-secretory phase of the menstrual cycle, which corresponds to the increasing level of progesterone and the presence of estrogen at the time of implantation [5]. Therefore, *HOXA10*/*Hoxa10* will be at its highest during the mid-to-late secretory phase of the menstrual cycle in healthy fertile women but not in women with endometriosis, which is believed to influence one’s fertility.

Endometriosis is a medical condition with cases ranging from mild to severe among women of various ages. Endometriosis occurs when endometrium-like tissues with the same functions as normal endometrial tissues are present outside the uterine cavity. Endometriosis is a chronic gynecological disorder related to estrogen that results in pelvic pain and infertility [6]. Approximately 176 million women in the world suffer infertility and severe pelvic pain due to endometriosis [7]. Women generally aged between 25 and 29 years old typically seek confirmation of endometriosis after complaining of pain in the pelvic area, while older women seek confirmation after the indication of infertility [8]. The origins of endometriosis remain unknown; however, some studies have suggested that endometriosis occurs due to retrograde menstruation, corresponding to Sampson’s hypothesis. Sampson’s theory explicating retrograde menstruation is a well-established and widely accepted explanation for the occurrence of endometriosis. It postulates that in this condition, menstrual blood, which comprises endometrial cells, reverses its flow through the fallopian tubes and enters the pelvic cavity, facilitating the implantation and growth of the cells. This theory has been supported by various studies and is an important concept in the understanding and treating of endometriosis [9]. There is a positive correlation between retrograde menstruation and endometriosis in which women with endometriosis have a higher prevalence of retrograde menstruation than women without endometriosis [10]. The classification of endometriosis can be divided by histopathology and anatomical locations into three subtypes: superficial endometriosis, deep infiltrating endometriosis, and ovarian endometriotic cysts [11].

Another possible pathogenesis of endometriosis is the occurrence of Müllerian duct anomalies (MDAs). Initially, the Wölffian (male) and Müllerian (female) ducts are formed during embryonic development via the intermediate mesoderm of the gastrula, and finally, one of the ducts will regress in order to determine the sex of the individual. In women, the development of the Müllerian duct is important as it will consequently differentiate into the different parts of the female reproductive tract, such as the fallopian tubes, uterus, cervix, and the superior aspect of the vagina [12]. Two transcription factors required for the differentiation and development of the Müllerian duct are *HOXA10* and homeobox 2 (*EMX2*). The expression levels of *HOXA10* and *EMX2* have been shown to decrease and increase consecutively in infertile women with MDAs [13]. Usually, women suffering from MDAs realize abnormalities in their female reproductive tract after puberty. Some realize the abnormalities due to pelvic pain, endometriosis, or infertility [14]. However, a diagnosis of an MDA can hardly be made because it is normally asymptomatic [15]. Endometriosis in women with MDAs is thought to be developed through retrograde menstruation specifically caused by outflow obstruction which subsequently floods the fallopian tubes and uterus, thus overwhelming the body’s ability to either remove or resorb and therefore resulting in endometriosis [14]. Apart from endometriosis, more diseases arise from the female reproductive tract, for example, salpingitis [16] and hydrosalpinx [17].

The occurrence of endometriosis in women eventually causes abnormal changes to gametes and embryos, eutopic endometrium, fallopian tubes, and embryo transport [18]. In addition, due to the inflammation caused by endometriosis, the endometrial function is disturbed due to changes in systemic and local cytokine expression [19]. Subsequently, this will disrupt endometrial receptivity. Research on endometriotic women has been performed for over a decade. However, the question of whether the expression of the *HOXA10* gene differs across populations regardless of geographic variations arises. Therefore, this systematic review aims to examine studies on the differential expression of *HOXA10* in women with and without endometriosis and the relationship between the *HOXA10* gene and endometriosis, which may interrupt fertility across the population. In addition, the evidence of fertility problems provided in some studies will be considered fertility problems commonly encountered by endometriotic women.

## 2. Methods

A systematic review of the publications was completed following the PRISMA guideline to identify studies that presented the differences in *HOXA10* gene expression in the endometria of women with and without endometriosis. The review protocol was registered with PROSPERO (CRD42022313381).

### 2.1. Criteria for Considering Studies for Review

#### 2.1.1. Types of Study

This systematic review included all observational studies for assessing *HOXA10* gene expression in infertile and fertile women with endometriosis compared to a control. In addition, human endometrial tissues from any endometriosis were included. Studies of HOXA10 protein expression and the hypermethylation of the *HOXA10* gene were excluded.

#### 2.1.2. Types of Participants

The participants were divided into patient and control groups. The patient group consisted of infertile and fertile women with endometriosis, while the control group consisted of women without endometriosis. The diagnosis of endometriosis in the patients was confirmed via endoscopy, ultrasonography, or histology. Other primates or species, such as baboons or mice, were excluded.

#### 2.1.3. Types of Intervention

The studies compared *HOXA10* gene expression levels between women with and without endometriosis and proved that fertility problems were included. No hormonal or fertility treatments were administered before sample collection.

#### 2.1.4. Types of Outcomes

Measurements of *HOXA10* expression levels during the implantation window in women with endometriosis were taken.

### 2.2. Search Strategy for the Identification of Studies

#### Electronic Searches

The keywords used were (HOXA10 OR “homeobox A10” OR PL OR HOX1 OR HOX1H OR HOX1.8) AND (“gene expression”) AND (endometriosis). The screening process is outlined as such (Figure 2). Conference abstracts, review articles, encyclopedias, and book chapters were excluded. No year restrictions were used.

### 2.3. Data Collection and Analysis

#### 2.3.1. Study Selection

All reviewers (NL, MFA, MHE, AU, SES, MHM, SZ, AKAK, and MAA) independently screened titles and abstracts against the inclusion criteria to identify eligible citations. NL and MHE found the full texts for all studies and decided if more studies needed to be excluded. Discussions including all reviewers resolved disagreements. NL wrote the first draft of the systematic review and reviewed it with others before submitting the final manuscript.

#### 2.3.2. Data Extraction

Data were extracted independently from the included studies by NL, SZ, MFA, and MHE. Disagreements were resolved via discussion. The author’s name, year, country, *HOXA10* expression methods, sample size, and age were extracted from all the studies. The type or localization of the endometriosis samples, fold change or *p*-value of *HOXA10* gene expression, and evidence of fertility problems were tabulated.

#### 2.3.3. Assessment of Risk of Bias

The risk of bias was assessed for each study using the NIH Quality Assessment Tool for Case-Control Studies [20]. All the studies were assigned a yes, no, or other to each of the 12 criteria allocated for each study and were evaluated as good, fair, or poor, depending on the rating made by the reviewers.

#### 2.3.4. Data Synthesis

Characteristics and main outcomes were tabulated for each study. Levels of *HOXA10* gene expression and evidence of fertility problems such as defective endometrial receptivity, implantation failure, low implantation rates, or poor pinopode development were observed in the selected studies.

## 3. Results and Discussion

Six hundred and twenty-three studies were identified via specified keywords. Twenty-eight duplicate studies were removed before the screening. Five hundred and ninety-five full-text articles were assessed for eligibility, and five hundred and eighty-five were excluded with reasons.

Ten articles were independently extracted based on the inclusion and exclusion criteria (Figure 2). A total of 455 participants, consisting of several groups such as infertile and fertile women with or without endometriosis, infertile women with different forms of endometriosis, women with or without endometriosis while excluding their fertility status, and fertile women without endometriosis, were included in the review.

### 3.1. Description of Studies

Nine studies used quantitative real-time polymerase chain reactions (qPCRs) to measure the expression levels of *HOXA10*, while one study used a ribonuclease assay (RPA). No participants received any medical or hormonal treatment before sample collection. Table 1 summarizes the characteristics of the included studies.

### 3.2. Assessment of Risk of Bias

All the studies included were rated as fair due to having 50–80% of the “yes” values assigned to each criterion, as stated in the NIH Quality Assessment Tool for Case-Control Studies (Appendix A).

### 3.3. Outcome Measures

Ten studies reported low levels of expression of *HOXA10* in women with endometriosis compared to women without endometriosis, as defined in Table 1. Five of the studies provided evidence of endometriosis-related fertility problems, while the other five only mentioned endometriosis-associated fertility problems and provided no evidence to support the statement (Table 2).

### 3.4. Discussions

Our systematic review found sufficient data to support that *HOXA10* gene expression levels are relatively low in women with endometriosis compared to women without endometriosis. All included studies provide similar primary outcomes regardless of population. In addition, 5 out of 10 studies offer evidence of fertility problems such as defects in endometrial receptivity, an unreceptive endometrium, the occurrence of implantation failure, and the prevalence of primary infertility. Therefore, a low level of expression of *HOXA10* is considered a reason for endometriosis-associated infertility. Two hundred and sixty-eight women with endometriosis demonstrated significantly low *HOXA10* expression levels compared to the control group. Low levels of HOXA10 expression are observed in samples of eutopic and ectopic endometria and in different types of endometrioses, for instance, deep infiltrating, ovarian, and superficial peritoneal endometrioses. This result strengthens the hypotheses in previous findings on the relationship between the expression of *HOXA10* in endometriotic women and fertility problems. The regulation of *HOXA10* is considered significant based on a *p* value < 0.5. However, in ectopic lesions, two studies provided opposite findings: one found a high level of *HOXA10* expression, while another found a low level of *HOXA10* expression.

Regardless of alterations in the expression of *HOXA10*, women with endometriosis can also be fertile. Hence, this suggests that fertility problems affecting endometrial receptivity, including implantation failure, may not be caused by the downregulation of *HOXA10*. Therefore, further investigation into other factors is needed to evaluate the underlying relationship between endometrial receptivity and endometriosis-related fertility. A study has shown that *HOXA10* expression does not cause a significant decrease even though the level of β_3_ integrin expression, which *HOXA10* regulates, is significantly decreased [22]. However, in another study, β_3_ integrin and *HOXA10* were proven to be co-expressed similarly in the endometrium. Altering *HOXA10* will cause corresponding alterations in β_3_ integrin, explaining the relation of *HOXA10* as a regulator of β_3_ integrin [32]. Other than β_3_ integrin, the level of expression of calcitonin mRNA in women with unexplained infertility is also reduced compared to fertile women [33]. The administration of calcitonin in endometrial cells demonstrated increased expression levels of other markers such as α_v_β_3_ integrins and LIF, which promotes blastocyst implantation and improves endometrial receptivity [34,35].

An assessment of endometrial receptivity markers, for instance, α_v_β_3_ integrins, leukemia inhibitory factor (LIF), and pinopodes, was performed in a study by Jana et al. to determine the factors crucial to successful implantation. All three markers suggested a reduction in blastocyst implantation as all of them were found to be significantly less expressed compared to controls. A positive correlation between the expression levels of *HOXA10* and α_v_β_3_ integrins and the poor development of pinopodes was observed in women with endometriosis compared to controls [23]. The presence of pinopodes can be considered a marker for endometrial receptivity because of their appearance during the luteal phase and their patterns of expression in the endometrium, which depend on the regulation of the individual’s hormones [36]. However, one study revealed that patients can become pregnant even when their pinopodes scored 0 (indicating that 0% of the apical surface was covered in pinopodes) in their previous cycle [37]. Due to these circumstances, additional evidence must be obtained by conducting detailed in vivo experiments to show the attachment of the blastocyst to the pinopodes during the implantation window to support the use of pinopodes as a marker for endometrial receptivity. Moreover, the development of pinopodes also appeared to have a connection with the levels of expression of thrombomodulin (TM) and ezrin, which affect the morphology of epithelial cells and the migration of collective cells [38]. Low levels of TM and ezrin have been discovered in recurrent pregnancy loss (RPL) compared to fertile controls; hence, inadequate pinopode development occurs due to the impaired regulation of TM and ezrin [39].

Hypotheses regarding the biomarkers of endometriosis have been published extensively in multiple journals; however, none of the experiments validated a single biomarker for noninvasive endometriosis tests [40]. ESHRE guidelines state that no biomarkers measured in serum or peritoneal fluid for the diagnosis of endometriosis are currently recommended [41]. Even so, noninvasive tests using a single biomarker or a panel of biomarkers aid in determining the pathogenesis of endometriosis and the development of endometriotic lesions. The most commonly investigated biomarkers are cancer antigen-125 (CA-125), cancer antigen-199 (CA-19-9), interleukin-6 (IL-6), and urocortin (UCN) [42,43]. Despite serving as an ovarian and endometrial cancer biomarker, CA-125 also acts as a potential biomarker for endometriosis. CA-125 levels are higher in women with endometriosis than controls [44]. The mean CA-125 level was reported to be 49.93 ± 4.30 U/mL, which is higher than the upper normal value, 35 U/mL [45]. However, this elevation is detected in endometriomas and deep infiltrating lesions but not in superficial peritoneal lesions [46]. A negative correlation between CA-125 and HOXA10 has been demonstrated, and elevation of CA-125 will possibly stimulate the progression of endometriosis through autophagy activation and HOXA10 deficiency [47]. Also, CA- 125 and CA-19-9 are useful in identifying the severity of endometriosis as the serum levels increase more when the rAFS score is higher [48,49]. Likewise, the increased secretion of cytokine IL-6 in peritoneal fluid (PF) and serum were attained in women with endometriosis [50], resulting in the modulation of Src homology region 2-containing protein tyrosine phosphatase-2 (SHP-2), which consequently suppressed natural killer (NK) cell activity [51]. Additionally, two out of three isoforms of UCN, UCN2 and UCN3, are less expressed in women with endometriosis compared to controls [52].

MicroRNAs (miRNAs) also play essential roles in controlling post-transcriptional gene expression. The dysregulation of miRNAs in endometriosis is still debatable as different studies use different types of samples; thus, the concordance of the studies within similar types of samples and the same miRNAs tested is limited. Most miRNAs, such as miR-197-5p, miR-22-3p, miR-320a, miR-320b, miR-3692-5p, miR-4476, miR-4520, miR-4532, miR-4721, miR-4758-5p, miR-494-3p, miR-6126, miR-6734-5p, miR-6776-5p, miR-6780b-5p, miR-6785-5p, miR-6791-5p, miR-939-5p, miR-22-3p, miR-125b, miR-150, miR-342, and miR-451a, are upregulated in women with endometriosis compared to controls [53,54,55], whereas miR-375, miR-139-5p [56], miR-200b, miR-15a-5p, miR-19b-1-5p, miR-146a-5p, and miR-200c [57] are downregulated. While they function as circulating biomarkers for disease risk and severity, they have been associated with the homeostasis of multiple biological systems as key regulators [58]. For example, the levels of circulating microRNA 135a (miR-135a) and microRNA 135b (miR-135b) were significantly increased in women with endometriosis, which caused the repression of *HOXA10* and consequently affected the function of the endometrium. Therefore, inverse relations of miR-135a/b and *HOXA10* can act as endometrial diagnostic and therapeutic biomarkers [25,59,60,61,62]. Other than that, the overexpression of miR-139-5p resulted in the downregulation of *HOXA10* and *HOXA9* [56]. The aberrant expression of miRNAs is also associated with various oncological diseases such as cervical cancer, lung cancer, prostate cancer, breast cancer, colorectal cancer, and gastric cancer [63,64,65,66,67,68].

A separate review assessing the role of *HOXA10* in endometriosis and infertility reported that *HOXA10* possibly influenced misplaced endometrial cells due to endometriotic lesions that localized near Müllerian structures [69]. It also stated that implantation failure is the primary cause of endometriosis-associated infertility. Two studies proposed that a different alteration in *HOXA10* occurs in women with varying forms of endometriosis [24,26]. In a study by Matsuzaki et al., a greater degree of alteration in *HOXA10* expression was found in patients with superficial peritoneal endometriosis (SE) compared to patients with deep infiltrating endometriosis (DE) or ovarian endometriosis (OE). A *p*-value of <0.001 was observed in SE patients, while a *p* value of <0.002 was observed in DE and OE patients. This study also indicated implantation failure in women with superficial peritoneal endometriosis but not in women with DE and OE. Thus, compared to OE, periods of infertility in women with SE are longer in duration, and their primary fertility and antral follicle counts (AFCs) are lower [70]. The downregulation of *HOXA10* in women with adenomyosis can also impact the rate of successful implantation [71].

The aberrant methylation of genes in the endometrium is known to cause abnormal endometrial function due to the atypical proliferation of endometrial cells. The silencing of *HOXA10* affected by the hypermethylation of its promoter region causes the downregulation of *HOXA10* and consequently impairs endometrial receptivity, thus preventing the development of a favorable environment for implantation [27,28]. Infertility in women with endometriosis had been associated with the abnormal regulation of *HOXA10* as levels of *HOXA10* were supposedly higher during the mid-secretory phase corresponding to the implantation window [72]. However, one study stated that the hypomethylation of *HOXA10* in women with endometriosis might be due to differences in population and interventions received compared to previous studies [73], while another study found the presence of and difference in the hypermethylation and hypomethylation of *HOXA10* depending on which sites of the CpG islands the methylation occurs and the type of endometriosis patients included in a study [74,75].

Furthermore, the sumoylation of HOXA10 by small ubiquitin like-modifier 1 (SUMO1) in women with recurrent implantation failure (RIF) revealed a negative effect on the embryo implantation process. The co-expression of HOXA10 and SUMO1 provides room for sumoylation to occur, and the overexpression of SUMO1 will increase HOXA10 ubiquitination. Through sumoylation, the inhibition of HOXA10 protein stability and transcriptional activity decreased the DNA-binding capacity of HOXA10, thus reducing HOXA10-mediated transactivation. The occurrence of sumoylation is best described as a the modification of a HOXA10 protein by SUMO1 on its evolutionarily conserved lysine 164 residue; hence, synthesized HOXA-SUMO1. HOXA10 sumoylation can be prevented through treatment with estradiol (E2) and progesterone (P). Research on the expression and function of small ubiquitin like-modifier (SUMO) in the mammalian endometrium is still limited, and HOXA10 sumoylation in endometriosis has not yet implemented [76].

Most studies state that endometriosis-associated infertility may be directly or indirectly affected by the downregulation of endometrial *HOXA10* gene expression; therefore, increasing its expression may improve one’s fertility. Endometrioma surgery is believed to assist in increasing the expression of Hox genes, including *HOXA10*, through the inhibition of *HOXA10* hypermethylation or the improvement of P resistance by reducing the responsive relationship between P and Hox genes. A significant fold change of 12.1 and a *p* value of 0.02 are achieved after endometrioma surgery; hence, it may improve one’s endometrial receptivity [77]. However, this study does not provide conclusive evidence for enhancing endometriosis-associated infertility following increased *HOXA10* expression. Other than endometriosis, endometrial hyperplasia also ought to occur due to various disruptions in the endometrium resulting from the loss of *HOXA10*, which can lead to endometrial cancer [78].

Recently, nanotechnology emerged as one of the promising non-invasive treatments for endometriosis. This technology has been well-established in cancer treatment and is thus being employed in endometriosis. Iron oxide-based magnetic nanoparticles (MNs), cobalt-coated and with irregular hexagonal shapes, were used to increase heating efficiency with the intention of elevating the intralesional temperature above 50 °C, consequently eradicating endometriotic lesions. The surfaces of the poly(ethylene glycol)-block-poly(ε-caprolactone) (PEG-PCL)-coated MNs were modified with peptides that targeted vascular endothelial growth factor receptor 2 (VEGFR-2/KDR) to enhance the accumulation and specificity towards endometriotic cells. Using nanoparticle-mediated magnetic hyperthermia can remove various sizes of endometriotic lesions. During the procedure, E2 and P must be sequentially administered to recapitulate the ovarian cycle [79]. Additionally, women experiencing endometriosis could also improve their condition by consuming metformin, letrozole, and dietary supplements. Metformin was reported to suppress endometriotic implant growth, reducing the expression of vascular endothelial growth factor receptor (VEGF) protein and mRNA while elevating the expression levels of LIF and HOXA10 [80]. Furthermore, letrozole increased the expression of αvβ3 and HOXA10 [81]. Due to the elevation of HOXA10, both drugs are believed to help improve endometrial receptivity in women with endometriosis. The consumption of dietary supplements is also one of the initiatives to complement the treatment of endometriosis. Some of these supplements are vitamin D, zinc, magnesium, omega 3, propolis, quercetin, curcumin, N-acetylcysteine, probiotics, resveratrol, alpha lipoic acid, vitamin C, vitamin E, selenium, and epigallocatechin-3-gallate [82].

Various treatments have been used worldwide to increase the probability of endometriotic women conceiving a child. The association of endometriosis with RIF and unsuccessful in vitro fertilization (IVF) is related to the state of the endometrium during the implantation window. Women with endometriosis that experienced RIF were reported to have negative outcomes during IVF treatment, as successful IVF depends on the success of embryo implantation. Despite this, the treated group who received treatment for endometriosis during IVF was deemed to have higher rates of pregnancy and live birth as the fertilization rate and available embryonic rate were higher than in the untreated group. The untreated group had a lower fertilization rate and available embryo rate, which were 46.24% and 85.20% compared to the treated group. In this study, the treated group controlled the activity of their endometriosis via treatment with a gonadotropin-releasing hormone agonist (GnRH-a) [83]. However, it remains a mystery whether the administration of GnRH-a, which resulted in positive outcomes in IVF treatment, helped due to changes in the expression of *HOXA10*, which is proposed to improve the condition of the endometrium during the implantation window.

## 4. Conclusions

This systematic review provides insight into the relationship between endometriosis and the expression of the *HOXA10* gene. Currently, sufficient data are provided to support the observation of low levels of *HOXA10* expression in women with endometriosis compared to women without endometriosis. Compared to mice, the expression of HOXA10 in humans had not been widely investigated; hence, fertility problems faced by endometriotic women still must be clarified, and no validated results have been achieved. This review provides similar data on *HOXA10* expression in endometriotic women across populations despite different geographical regions, which consequently provides an innovative idea for closing the research gap for infertility treatment. Predictive tools such as biomarkers are essential in determining the presence and severity of endometriosis. Further research is needed to ascertain the underlying mechanisms of endometriosis’s pathogenesis and achieve conclusive results in endometriosis-associated fertility.

## Figures and Tables

**Figure 1 ijms-24-12869-f001:**
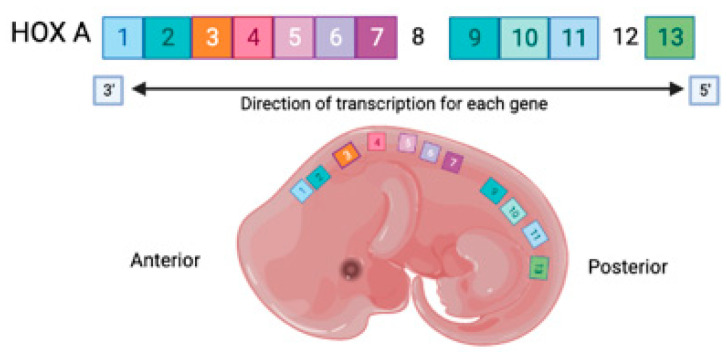
The mammalian HOX/Hox A gene clusters and their regional expression during embryonic development. The color-coding of *HOXA/Hoxa* 1–13 (except for 8 and 12) represents the colinear expression of the HOX A gene at different regions.

**Figure 2 ijms-24-12869-f002:**
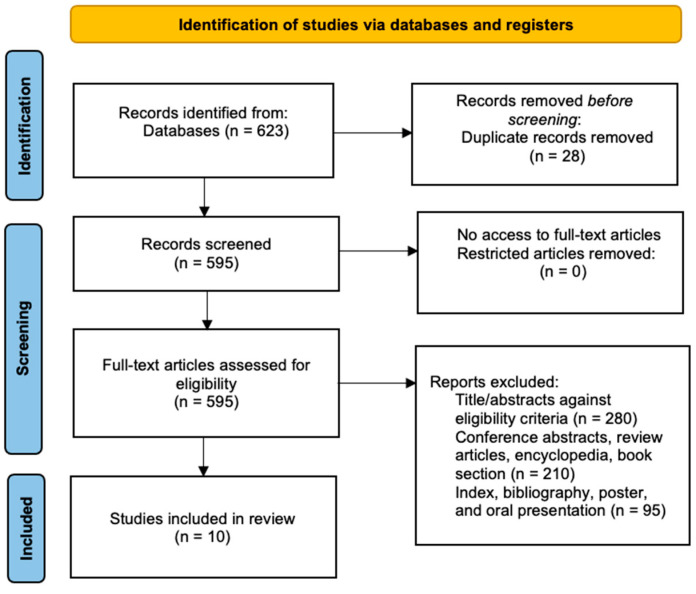
PRISMA flow diagram for the screening process in this systematic review [21].

**Table 1 ijms-24-12869-t001:** Characteristics of included studies.

No	Author	Year	Country	Method (*HOXA10* Expression)	Participants	Age (Years Old)
1	Gui et al. [22]	1999	USA	Ribonuclease protection assay (RPA)	Endometriotic women (*n* = 41) Control (*n* = 35)	N/A
2	Jana et al. [23]	2013	India	Quantitative PCR (qPCR)	Infertile endometriotic women (*n* = 31) Control (*n* = 26)	<35
3	Matsuzaki et al. [24]	2009	France	qPCR	Infertile women with different forms of endometriosis (*n* = 62) Control (*n* = 20)	<38
4	Mirabutalebi et al. [25]	2018	Iran	qPCR	Endometriotic women (*n* = 34) Control (*n* = 17)	20–45
5	Özcan et al. [26]	2019	Turkey	qPCR	Infertile endometriotic women (*n* = 11) Fertile endometriotic women (*n* = 11) Control (*n* = 11)	≤39
6	Szczepánska et al. [27]	2010	Poland	qPCR	Infertile endometriotic women (*n* = 17) Control (*n* = 15)	25–39
7	Wu et al. [28]	2005	USA	qPCR	Endometriotic women (*n* = 6) Control (*n* = 4)	26–38
8	Lu et al. [29]	2013	China	qPCR	Endometriotic women (*n* = 6) Control (*n* = 6)	N/A
9	Samadieh et al. [30]	2019	Iran	qPCR	Endometriotic women (*n* = 36) Control (*n* = 21)	20–40
10	Wang et al. [31]	2018	China	qPCR	Endometriotic women (*n* = 30) Control (*n* = 15)	25–37

**Table 2 ijms-24-12869-t002:** Outcome data of included studies on endometriosis patients.

Author	HOXA10 Gene Expression	Type of Endometriosis Samples	Fold Change	*p*-Value	Evidence of Fertility Problems
Gui et al. [22]	Downregulated	N/A	N/A	N/A	Defects in endometrial receptivity
Jana et al. [23]	Downregulated	N/A	N/A	N/A	Endometrial receptivity in an unreceptive state with poor pinopode development
Matsuzaki et al. [24]	Downregulated	Deep infiltrating ^b^ Ovarian ^b^ Superficial peritoneal ^b^	N/A N/A N/A	<0.001 <0.002 <0.002	Occurrence of implantation failure
Mirabutalebi et al. [25]	Downregulated Upregulated	Eutopic endometrium ^c^ Ectopic lesions ^c^	N/A N/A	0.001 0.681	N/A
Özcan et al. [26]	Downregulated	Ovarian ^a^ Ovarian ^b^	1871 3509	N/A N/A	N/A
Szczepánska et al. [27]	Downregulated	Eutopic endometrium ^b^	N/A	0.019	N/A
Wu et al. [28]	Downregulated	N/A	N/A	N/A	Defects in endometrial receptivity
Lu et al. [29]	Downregulated	Eutopic endometrium ^a^	N/A	<0.05	N/A
Samadieh et al. [30]	Downregulated	Ovarian ^b^	N/A	N/A	N/A
Wang et al. [31]	Downregulated	Eutopic endometrium ^b^ Ectopic lesions ^b^	N/A	<0.001 <0.001	Prevalence of primary infertility

^a^ Samples from fertile endometriotic women. ^b^ Samples from infertile endometriotic women. ^c^ Samples from women with unknown fertility.

## Data Availability

Not applicable.

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
