# Peer review of "Expression of HOXA10 Gene in Women with Endometriosis: A Systematic Review"

_ijms, 2023, doi:10.3390/ijms241612869_

Round 1

Reviewer 1 Report

Comments and Suggestions for Authors

In the present report, Lazim et al nicely survey a rigorous body of prior reports concerning the expression and potential relevance of HOXA10 expression as it relates to the development and severity of endometriosis.  The paper is well-written in general and organized appropriately.  There are minor suggestions for improvement.

There are a few cases where there is a conclusion like "may or may not be important" this is true for any random gene as it may or may not be relevant to endometriosis.  I'd eliminate wishy-washy conclusions and just state something like "While not entirely clear from these reports, most of the evidence is for/against X" etc.

Figure 1 is not adequately described, particularly in the legend, and I'm not sure what the point of it is.  I assume it's showing colinear expression (with the top part being gene order on the chromosome) and spatial or temporal regulation of the genes in body segmentation or CNS development.  The legend mentions conserved signaling pathways, but I don't see any signaling pathways.  Further, there is only one embryo shown so what about this indicates conservation in more than one mammalian species.

The introduction is sufficiently detailed.  However, around L60 there is a discussion about endometriosis being diagnosed by different symptoms at different ages.  However, as far as I know, the only "diagnosis" is via verification of endometriotic lesions from laparoscopy.  You could temper this by saying women at age X typically seek confirmation of endometriosis if they exhibit Y, etc.

I think it's generally accepted that retrograde menstruation is the cause of endometriosis.

L171, clarify that it's RT-PCR (not just PCR) that was likely used.  Also, an indication of how much was quantitative analysis such as real-time RT-PCR vs semi-quantitative might be relevant as well.

A nice panel of putative HOXA10-regulated factors is presented in the discussion with suitable evidence of links to endometriosis-related infertility.

Author Response

[General Comments] Lazim et al nicely survey a rigorous body of prior reports concerning the expression and potential relevance of HOXA10 expression as it relates to the development and severity of endometriosis.

Response: We thank the reviewer for the positive comment.

[Minor Comment 1] There are a few cases where there is a conclusion like "may or may not be important" this is true for any random gene as it may or may not be relevant to endometriosis.  I'd eliminate wishy-washy conclusions and just state something like "While not entirely clear from these reports, most of the evidence is for/against X" etc.

Response: We revised the sentence as follows:

[L109]

 “Therefore, this systematic review aims to examine the studies on the differential expression of HOXA10 in women with and without endometriosis and the relationship between the HOXA10 gene and endometriosis, which may interrupt one’s fertility across the population.”

[L218]

“Hence, this suggests that fertility problems affecting endometrial receptivity, including implantation failure, may not be caused by the downregulation of HOXA10.”

[Minor Comment 2] Figure 1 is not adequately described, particularly in the legend, and I'm not sure what the point of it is. I assume it's showing colinear expression (with the top part being gene order on the chromosome) and spatial or temporal regulation of the genes in body segmentation or CNS development.  The legend mentions conserved signaling pathways, but I don't see any signaling pathways.  Further, there is only one embryo shown so what about this indicates conservation in more than one mammalian species.

Response: We appreciate you bringing that to our attention. We apologize for any confusion that may have occurred. This figure is actually not about signaling pathways but about embryogenesis. And yes, it’s showing colinear expression of the HOX A gene at different regions. Therefore, we revised the paragraph and the legend as follows:

[L38]

“Hox genes are involved in the development of the female reproductive tract during embryogenesis. The HOX A gene clusters are regionally expressed during embryonic development in mammals such as mice and humans (Figure 1). Specifically, expression of HOXA10/Hoxa10 is observed in the developing uterus, while Hoxa11 is observed in both the developing uterus and the cervix.”

[L56]

“Figure 1. The mammalian HOX/Hox A gene clusters and regional expression during embryonic development. Color-coding of HOXA/Hoxa 1-13 (except 8 and 12) represented the colinear expression of the HOX A gene at different regions.”

[Minor Comment 3] The introduction is sufficiently detailed.  However, around L60 there is a discussion about endometriosis being diagnosed by different symptoms at different ages.  However, as far as I know, the only "diagnosis" is via verification of endometriotic lesions from laparoscopy.  You could temper this by saying women at age X typically seek confirmation of endometriosis if they exhibit Y, etc.

Response: We agree with you. Pelvic pain and infertility are not diagnoses but rather symptoms. Thus, we revised the sentence as follows:

[L66]

“Women generally aged between 25 to 29 years old typically seek confirmation of endometriosis after complaining of pain in the pelvic area, while older women seek confirmation after indication of infertility.”

[Minor Comment 4] I think it's generally accepted that retrograde menstruation is the cause of endometriosis.

Response: Thank you for pointing that out. Based on a journal we came across, retrograde menstruation is widely accepted as the cause of endometriosis. Therefore, we revised the paragraph as follows:

[L71]

“Sampson's theory explicating retrograde menstruation is a well-established and widely accepted explanation for the occurrence of endometriosis. It postulates that in this condition, menstrual blood, which comprises endometrial cells, reverses its flow through the fallopian tubes and enters the pelvic cavity, facilitating implantation and growth of the cells. This theory has been supported by various studies and is an important concept in the understanding and treating of endometriosis.”

[Minor Comment 5] L171, clarify that it's RT-PCR (not just PCR) that was likely used. 

Response: We revised the sentence as follows:

[L182]

“Nine studies used quantitative real-time polymerase chain reaction (qPCR) to measure the expression of HOXA10, while one used ribonuclease assay (RPA).”

[Minor Comment 6] Also, an indication of how much was quantitative analysis such as real-time RT-PCR vs semi-quantitative might be relevant as well.

Response: L197, an indication of how much quantitative analysis (qPCR) is tabulated in Table 2 in column “HOXA10expression”.  L186, to detect the HOXA10 gene, the method used is stated in Table 1. To avoid confusion to the reader on what this method is used for, we revised column “Method” to “Method (HOXA10 expression)”.

[General Comments] A nice panel of putative HOXA10-regulated factors is presented in the discussion with suitable evidence of links to endometriosis-related infertility.

Response: We thank the reviewer for the positive comment.

Reviewer 2 Report

Comments and Suggestions for Authors

HOXA10/Hoxa10 will be at its highest during the mid to late secretory phase of the menstrual cycle in healthy fertile women but not in women with endometriosis, which is believed to influence one’s fertility. Low expression of HOXA10 is considered a reason for endometriosis-associated infertility. Regardless of alteration in HOXA10 expression, women with endometriosis can also  be fertile. Hence, this suggests that fertility problems affecting endometrial receptivity,  including implantation failure, may or may not be caused by the downregulation of HOXA10

comments

(CA-199), line 246 please correct for 19-9

The Authors should cite other situations associated with a decreased expression of HOXA 10  

For example

-HOXA10 gene expression is decreased in the secretory phase endometrium of women with adenomyosis. Diminished expression of HOXA10 is a potential mechanism explaining decreased implantation observed in women with adenomyosis. Fertil Steril ,2011 Mar 1;95(3):1133-6

-decreased expression of HOXA10 in the decidua after embrio implantation promotes  trophoblast invasion Endocrinology . 2017 Aug 1;158(8):2618-2633.

This review should stigmatize that downregulation of HOXA 10 may be a marker of endometriosis but also of other situations and therefore should be ruled out as being involved in the pathogenesis of this disease

Author Response

[Minor Comment 1] (CA-199), line 246 please correct for 19-9

Response: Thank you for pointing that out. We revised the sentence as follows:

[L256]

“The most investigated biomarkers are cancer antigen-125 (CA-125), cancer antigen-199 (CA-19-9), interleukin-6 (IL-6), and urocortin (UCN).”

[L266]

“Also, CA- 125 and CA-19-9 are useful in identifying the severity of endometriosis as the serum levels arise more when the rAFS score higher.”

[Minor Comment 2] The Authors should cite other situations associated with a decreased expression of HOXA 10.

For example

-HOXA10 gene expression is decreased in the secretory phase endometrium of women with adenomyosis. Diminished expression of HOXA10 is a potential mechanism explaining decreased implantation observed in women with adenomyosis. Fertil Steril ,2011 Mar 1;95(3):1133-6

-decreased expression of HOXA10 in the decidua after embrio implantation promotes  trophoblast invasion Endocrinology . 2017 Aug 1;158(8):2618-2633.

This review should stigmatize that downregulation of HOXA 10 may be a marker of endometriosis but also of other situations and therefore should be ruled out as being involved in the pathogenesis of this disease

Response: Thank you for the suggestions. We can include the first article, but not the second one, as the data came from healthy women. HOXA10 expression alone cannot be used as a biomarker for endometriosis; however, this doesn’t mean aberrant regulation of HOXA10 is not involved in the pathogenesis of endometriosis. We added the first article into a paragraph as follows:

[L306]

“The downregulation of HOXA10 in women with adenomyosis can also impact the rate of successful implantation.”